# Learning Discrete Directed Acyclic Graphs via Backpropagation

**Andrew J. Wren**
University College London
andrew.wren@ntlworld.com

**Pasquale Minervini**
The University of Edinburgh
University College London
p.minervini@ed.ac.uk

**Luca Franceschi**
Amazon Web Services
franuluc@amazon.de

**Valentina Zantedeschi**
ServiceNow Research
University College London
INRIA London

## Abstract

Recently continuous relaxations have been proposed in order to learn Directed Acyclic Graphs (DAGs) from data by backpropagation, instead of using combinatorial optimization. However, a number of techniques for fully discrete backpropagation could instead be applied. In this paper, we explore that direction and propose DAG-DB, a framework for learning DAGs by Discrete Backpropagation. Based on the architecture of Implicit Maximum Likelihood Estimation [I-MLE, 1], DAG-DB adopts a probabilistic approach to the problem, sampling binary adjacency matrices from an implicit probability distribution. DAG-DB learns a parameter for the distribution from the loss incurred by each sample, performing competitively using either of two fully discrete backpropagation techniques, namely I-MLE and Straight-Through Estimation.

## 1 Introduction

**Aim.** Directed Acyclic Graphs (DAGs) occur in a wide range of contexts, including project management, version control systems, evolutionary biology, and Bayesian networks. Bayesian networks have been a particularly popular subject for machine learning, including for the problem considered in this paper, learning a Bayesian network's DAG structure from data. This paper aims to learn DAGs using fully-discrete backpropagation, i.e. avoiding continuous relaxations, which are usually used for learning DAGs from data – for example, see [2–13].

**Bayesian networks.** A Bayesian network associates a DAG with a random variable which has components indexed by the DAG nodes. A directed edge $i \longrightarrow j$ between nodes $i$ and $j$ represents a dependency of the random variable's $j$ component on its $i$ component. Discussion of Bayesian networks and DAGs may be found in textbooks (e.g., see [14–16]). To establish our terminology, appendix A gives a very brief outline.

**Contribution.** Our contribution is to show that backpropagation methods which retain the fully-discrete nature of a DAG (i.e., that do not rely on continuous relaxations) can be used to predict DAGs from data. We avoid the common approach of "relaxing" digraph adjacency matrices from binary to real matrices. Instead, we sample discrete variables probabilistically and use methods of backpropagating that do not relax the variables to be continuous.

36th Conference on Neural Information Processing Systems (NeurIPS 2022).

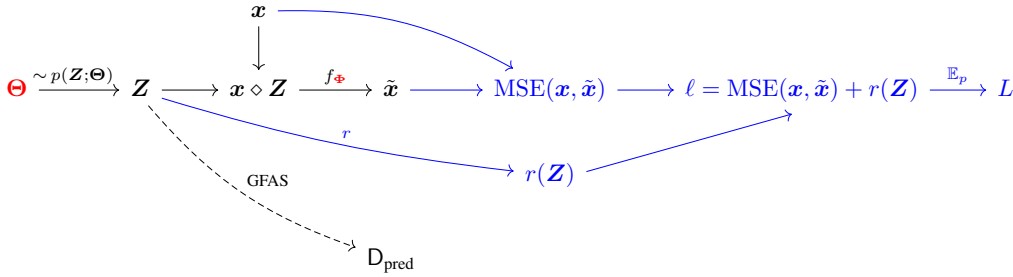

Figure 1: Overall DAG-DB architecture, including learnable parameters and loss calculation.

## 2 Related work and background

**Learning DAGs from data.** Most ways of learning DAGs from data can be classed as combinatoric methods (see [17] for a review) or continuous optimisation methods (reviewed in [18]). Combinatoric methods may themselves be labelled as constraint-based, identifying the graph from conditional independence testing, and score-based, searching the space of possible graphs using a score function to evaluate search results [17]. The PC algorithm [19] is a well-known constraint-based method, with a number of variant algorithms, such as PC-Stable [20], which reduces dependency on arbitrary node ordering. An example of a score-based approach is the Fast Greedy Equivalence Search (FGES) algorithm [21], which is an optimised version of the Greedy Equivalence Search (GES) [22, 23]. The development of NOTEARS in 2018 [2] for linearly-generated data signalled the start of widespread development of continuous optimisation approaches, including the GOLEM method [8] which is generally even more successful than NOTEARS for linear data with, for example, Gaussian noise. Many other continuous methods have recently been developed (e.g., see [3–7, 9–13]) and the method presented in this paper, which uses gradient descent, may be viewed as related to these. However, continuous methods generally adopt some approach of 'relaxing' the discrete property of an edge being present or absent into a continuous variable, whereas our DAG-DB approach maintains edges as fully discrete, binary objects, even during our method's training phase. Our approach might be termed a 'probabilistic relaxation', and another example of this can be found in the SDI [24] framework, which focuses on contexts where data may be generated by interventions during training, whereas, here, our focus is on pre-generated 'observational' data. Recently, a Bayesian approach has also been taken (e.g., see [25–27]), focusing on predicting a distribution of DAGs, instead of, as we do, seeking to predict only the most likely.

**Discrete backpropagation.** A number of methods have also been developed to allow backprop-agation without transforming discrete variables to be continuous in training. We take four such methods as examples. Straight-through estimation [STE, 28, 29] ignores the derivative of a discrete function during back-propagation and passes on the incoming gradient as if the function was the identity function; this simple approach, as we will see, can be surprisingly effective. Score-function estimation [SFE, 30], also referred to as REINFORCE, rewrites the gradient of an expectation in an expectation using the log-derivative trick, which can be then estimated using Monte Carlo methods. Black-box differentiation [BB, 31] is a way of adjusting continuous inputs to a combinatorial solver in order to mimic gradient descent for the discrete variable. Implicit maximum likelihood estimation [I-MLE, 1] incorporates noise into BB to handle discrete random variables. This allows combinatorial solvers to be used for maximum likelihood estimation. I-MLE also makes use of both STE and a seminal result by Domke [32]. We note that SDI [24], highlighted above, uses a particular Bernoulli-based SFE-like form of backpropagation [33].

**Perturb-and-MAP sampling.** We use the technique of perturb-and-MAP (P&M) sampling of random variables [34]. This approximates a matrix distribution, say $p(\boldsymbol{Z}; \boldsymbol{\Theta})$, with parameter $\boldsymbol{\Theta}$, as

$$p(\boldsymbol{Z}; \boldsymbol{\Theta}) \sim \mathrm{MAP}(\boldsymbol{\Theta} + \tau \boldsymbol{\Psi}), \tag{1}$$

where each element of the noise matrix $\boldsymbol{\Psi}$ is i.i.d. and sampled from a standard one-dimensional distribution, $\tau > 0$ is a temperature, and $\mathrm{MAP}(\boldsymbol{\Theta} + \tau \boldsymbol{\Psi})$ denotes the most likely value of $\boldsymbol{Z}$ with

respect to $p(\boldsymbol{Z}; \boldsymbol{\Theta} + \tau\boldsymbol{\Psi})$. P&M is useful when sampling $p(\boldsymbol{Z}; \boldsymbol{\Theta})$ directly is intractable or expensive, and the MAP solver is cheap, and facilitates I-MLE's approach to backpropagation.

## 3  Method

**Framework.**  Figure 1 gives an overview of the DAG-DB framework. Suppose we wish to learn a DAG with $d$ nodes from a dataset $\boldsymbol{X} \in \mathbb{R}^{n \times d}$ of $n$ data points $\boldsymbol{x} \in \mathbb{R}^d$. Let $\mathcal{R} \subset \mathbb{R}^{d \times d}$ and $\mathcal{B} \subset \{0, 1\}^{d \times d}$ be the sets of zero-diagonal, respectively real and binary, matrices. A learnable vector $\boldsymbol{\Theta} \in \mathcal{R}$ parameterises a exponential family distribution $p(\boldsymbol{Z}; \boldsymbol{\Theta})$,

$$p(\boldsymbol{Z}; \boldsymbol{\Theta}) = \exp\left(\langle \boldsymbol{Z}, \boldsymbol{\Theta}\rangle_{\mathrm{F}}/\tau - \mathrm{A}(\boldsymbol{\Theta})\right), \tag{2}$$

where $\boldsymbol{Z} \in \mathcal{Z} \subseteq \mathcal{B}$ is a discrete matrix, $\tau > 0$ is a temperature, and $\mathrm{A}(\boldsymbol{\Theta})$ normalizes $p(\boldsymbol{Z}; \boldsymbol{\Theta})$ to sum to one. Recognising the matrix form of $\boldsymbol{\Theta}$ and $\boldsymbol{Z}$, eq. (2) uses the Frobenius scalar product,

$$\langle \boldsymbol{Z}, \boldsymbol{\Theta}\rangle_{\mathrm{F}} \coloneqq \mathrm{tr}\left(\boldsymbol{Z}^{\mathsf{T}}\boldsymbol{\Theta}\right) = \sum_{\mathrm{ij}} \boldsymbol{Z}_{\mathrm{ij}}\boldsymbol{\Theta}_{\mathrm{ij}}. \tag{3}$$

The matrix $\boldsymbol{Z}$ is interpreted as the adjacency matrix of a directed graph on $d$ nodes, with, because of its zero diagonal, no self-loops. In the forward pass, samples $\boldsymbol{Z}^{(s)}$, $s = 1, ..., S$, from $p(\boldsymbol{Z}; \boldsymbol{\Theta})$ are taken using the P&M sampling outlined in section 2. For $\mathcal{Z} = \mathcal{B}$, the MAP solver sets matrix elements of $\mathrm{MAP}(\boldsymbol{\Theta})$ to be one if $\boldsymbol{\Theta} > 0$ and zero otherwise. We found empirically that it is best to use the standard logistic distribution to generate the noise $\boldsymbol{\Psi}$ in eq. (1). This can also be theoretically justified as a good choice because of the binary nature of $\boldsymbol{Z}$'s matrix elements [34].

**Maximum DAG.**  To go from the parameter $\boldsymbol{\Theta}$ and a digraph $\boldsymbol{Z}$ to a DAG, we can consider the digraph edges as weighted by the corresponding matrix elements of $\boldsymbol{\Theta}$, and then solve the associated maximum directed acyclic subgraph problem. Solving this well-known "maximum DAG" problem involves identifying the directed acyclic sub-graph with the maximum sum of weights. To find the maximum DAG from $\boldsymbol{\Theta}$ and $\boldsymbol{Z}$, we devise both exact and approximate solvers. For exactly solving the maximum DAG problem, we cast it as a constrained combinatorial optimisation problem, which we solve using MiniZinc [35, 36], a free and open-source constraint modelling language. We found it most efficient, and necessary as the number of digraph nodes rose toward 100, to use an approximate maximum DAG solver, "Greedy Feedback Arc Set" [GFAS, 37], our Python implementation being adapted from (more optimised) Java code [38, 39]. Tests for 30 nodes and fewer, for which we could use the exact MiniZinc solver, suggested that the solutions of our approximate GFAS solver were in practice exact, or close to exact, due, most likely, to regularizing (see below) $\boldsymbol{Z}$ towards being a DAG. Whatever maximum DAG solver is chosen, it can be applied either directly as part of the MAP solver, or subsequent to the MAP solver. We adopt latter option, which has the advantage of allowing an approach in which training is done without the maximum DAG solver, deploying it only at evaluation. In all but one part of an experiment, that 'evaluation only' approach is the one we will use.

**Learning**  As is common for continuous optimisation methods, learning proceeds by solving the problem of predicting values of $\boldsymbol{x}$ components $\boldsymbol{x}_{\mathrm{j}}$ from its values $\boldsymbol{x}_{\mathrm{i}}$ at 'parent' nodes with edges $\mathrm{i} \longrightarrow \mathrm{j}$. As set out in appendix C, this is ensured by the *graphification* operation $\boldsymbol{x} \diamond \boldsymbol{Z}$ in the context of the linear map $f_{\boldsymbol{\Phi}}$. Elsewhere, this technique usually employs a weighted adjacency matrix with real elements (see e.g., [2, 6, 8]); but we use the binary adjacency matrix $\boldsymbol{Z}$ without such a relaxation. To the extent we relax the problem, this is by treating the digraph probabilistically via the distribution $p(\boldsymbol{Z}; \boldsymbol{\Theta})$, which, in practice, becomes concentrated around the most likely adjacency matrix.

The parameters $\boldsymbol{\Theta}$ and $\boldsymbol{\Phi}$ are trained by feeding data points $\boldsymbol{x} \in \boldsymbol{X}$ into the model batch-wise. The mean-squared error between $\tilde{\boldsymbol{x}}$ and $\boldsymbol{x}$ is added to a regularizing function $r(\boldsymbol{Z})$ to give a loss $\ell$ associated with the sample $\boldsymbol{Z}$. The empirical mean of these losses, over the samples $\boldsymbol{Z}$ and the batch members $\boldsymbol{x}$, gives the batch loss $L$. Our learnt parameters $\boldsymbol{\Phi}$ and $\boldsymbol{\Theta}$ are then updated by backpropagation: while for $\boldsymbol{\Phi}$ this is standard backpropagation, a technique needs to be chosen for discrete backpropagation from $\boldsymbol{Z}$ to $\boldsymbol{\Theta}$. We found I-MLE [1] and straight-through estimation [28, 29] to be useful techniques for such backpropagation; we had less success here with score-function estimation [30] and black-box differentiation [31]. Appendix B gives details of I-MLE and straight-through estimation as used in DAG-DB.

**Regularization.** DAG-DB can employ both functional and constraint regularization. Functional regularization is based on that of NOTEARS [2] and provided by the function

$$r(\boldsymbol{Z}) = \rho_{\text{DAG}}\, r_{\text{DAG}}(\boldsymbol{Z}) + \rho_{\text{sp}}\, r_{\text{sp}}(\boldsymbol{Z}), \tag{4}$$

where $\rho_{\text{DAG}}, \rho_{\text{sp}} > 0$ are strength coefficients. For the DAG-regularizer $r_{\text{DAG}}$, we use the binary version of the DAG regularize used by NOTEARS,

$$r_{\text{DAG}}(\boldsymbol{Z}) = \left[\text{tr}\left(\exp \boldsymbol{Z}\right) - d\right]^2, \tag{5}$$

which has been shown [2, prop. 2] to vanish if and only if $\boldsymbol{Z}$ is the adjacency matrix for a DAG. The second regularizer, $r_{\text{sp}}$, is an $L^1$-regularizer promoting sparsity by summing $\boldsymbol{Z}$'s matrix elements. The other form of regularization is to constrain $\boldsymbol{Z}$ to within a proper subset $\mathcal{Z} \subsetneq \mathcal{B}$. We optionally introduce a maximum size (i.e. number of edges) of digraph, constraining $\boldsymbol{Z}$ to have no more than a given maximum number $M$ of non-zero entries (edges). For a maximum size $M$, the MAP solver identifies the $M$ biggest elements of $\boldsymbol{\Theta}$, but drops any that are not positive. Matrix elements of $\text{MAP}(\boldsymbol{\Theta})$ corresponding to the resulting index set have value one, and the remainder have value zero.

## 4 Experiments

**Overview.** As detailed in appendix D.3, we identified useful hyperparameters settings, `STE_84`, using STE with a maximum size 84, and `IMLE_None`, using I-MLE with no maximum size constraint. We then conducted ablation experiments, and tests on linearly-generated synthetic data, and on a real dataset. The main metric we use is $\text{SHD}_{\text{c}}$, often normalised to $\text{nSHD}_{\text{c}}$, complemented by a precision measure $\text{prec}_{\text{c}}$ and, sometimes also by a recall measure $\text{rec}_{\text{c}}$. Appendix D.2 gives more details on these metrics, as part of supplementary material on the experiments in appendix D.

**Ablation experiments.** We performed further ablation experiments to identify the relative role of the regularizers, with results shown in appendix D.4. We found that $r_{\text{DAG}}$, and, the appropriate setting or not of a maximum size regularizer, both have a notable good effect. However, the presence of a sparsity regularizer $r_{\text{sp}}$, although almost always helpful, was only very relevant for `STE_84`, and then only when the maximum size constraint was also ablated.

**Synthetic data experiments.** We generated random Erdös-Rényi 'ER$k$' and Barabási-Albert 'SF$k$' DAGs and further generated data from these using a Gaussian equal-variance linear additive noise model as the Bayesian network. The number of edges possible in these synthetic graphs varies considerably, so the maximum size limit for `STE_84` was adjusted in proportion to the expected size. We tested our methods against those highlighted in section 2 on sets of 24 random graphs, with results shown in fig. 2. We can see that the DAG-DB methods usually outperform the combinatorial approaches, and they are themselves outperformed by the other continuous methods. Of our methods, `STE_84` performs better, with the maximum size adjustments noted above, while `IMLE_None`'s performance is fairly close, except for ER4 and SF4 with $d = 100$ nodes, where its metrics worsen noticeably.

**Real data experiments.** We performed experiments on the Sachs cellular biochemistry dataset, where each instance can be represented as a DAG with 11 nodes and 17 edges [40]. Performance noted previously [2, 8] suggests that linear models perform reasonably on Sachs' 853-data point set of purely observational data. Table 1 compares our results against other methods. For `STE_84`, the table also indicates maximum size constraints which have to be reset, by a proportioning method (23) and with reference to typical, relatively successful, sizes predicted by other methods (8). We can see that `IMLE_None` is much more successful than either `STE_84` setting, and, to account for a stochastic element in DAG-DB results, we therefore show, for I-MLE only, the results over 24 runs on the same Sachs data. We can see from the table that `IMLE_None`'s performance is quite close to that of GOLEM and NOTEARS which are the best of the comparison methods. Standard deviations are discussed in appendix D.7.

We also ran a method `IMLE_None_Tr`, which, unlike our other DAG-DB methods, applies GFAS to make $\boldsymbol{Z}$ the adjacency matrix of a DAG, *in training*. In light of the associated compute time, we did not undertake a full hyperparameter search for `IMLE_None`, and instead only vary the $\lambda$ hyperparameter to which I-MLE can be sensitive. Generally digraph training performs marginally better on $\text{SHD}_{\text{c}}$ and often markedly better on $\text{prec}_{\text{c}}$ and $\text{rec}_{\text{c}}$. This may be because it was not practical to optimise hyperparameters for DAG training. The results can be found in appendix D.7.

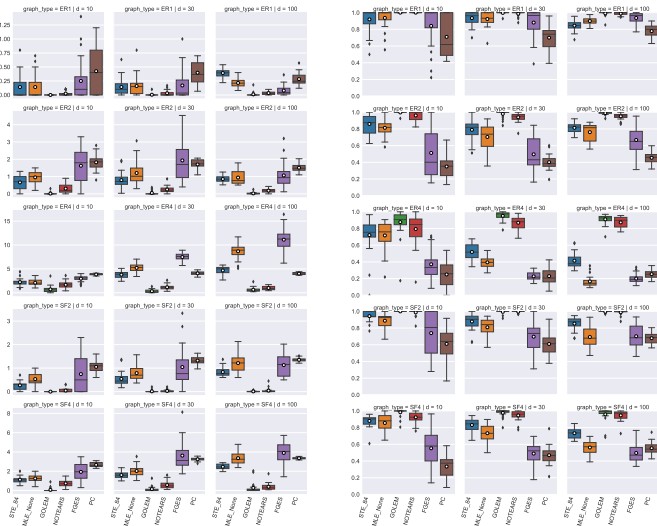

Figure 2: Tests of eight methods against selected graph type *(rows)* and numbers of nodes $d$ *(columns)*. The two panels show $\mathrm{nSHD_c}$ *(left)* for which low values are good, and $\mathrm{prec_c}$ *(right)* for which high values are good. Vertical axis scales vary by panel, and, in the left-hand panel, also by row. The box-and-whiskers plots indicate spread over 24 test DAGs (see appendix D.5 for details).

Table 1: Metrics for selected models for the Sachs [40] observational dataset of 853 data points. Best metric scores are in bold. See the text and appendices for discussion.

| Model | | $\mathrm{SHD_c}$ | $\mathrm{prec_c}$ | $\mathrm{rec_c}$ | pred. size |
|---|---|---|---|---|---|
| STE_84 | MAX_SIZE $= 23$ | 20 | 0.158 | 0.176 | 19 |
| | MAX_SIZE $= \phantom{0}8$ | 15 | 0.600 | 0.176 | 5 |
| IMLE_None | mean | 12.7 | 0.869 | 0.255 | 5.0 |
| | median | 13 | **1.000** | 0.235 | 5 |
| GOLEM | | **11** | **1.000** | 0.353 | 6 |
| NOTEARS | | **11** | 0.467 | **0.412** | 15 |
| FGES | | **11** | 0.750 | 0.353 | 8 |
| PC | | **11** | 0.750 | 0.353 | 8 |

# 5   Conclusion and future work

For linear data, DAG-DB mostly outperforms combinatorial methods tested here, but is itself outperformed by the continuous methods. The DAG-DB framework should adapt to data generated by non-linear models and, potentially, to discrete data, or to causal models [15]. It would also be interesting to see if the SFE-like backpropagation [33] used in SDI [24] would work for DAG-DB. Separately, one of the I-MLE paper's experiments [1] built a variational auto-encoder (VAE), and a similar experiment could be performed with a latent space of DAGs. A VAE might learn a single DAG for a given dataset, and also learn values for each data point on that DAG's nodes. This might be interpretable as learning a discrete hierarchy representing characteristics of the dataset.

## Acknowledgments and Disclosure of Funding

Andrew J. Wren is very grateful to his co-authors who supervised him in the University College London MSc Machine Learning thesis project from which this paper derives.

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

# A  Notation and terminology

**Notation.**  We write vectors in bold, like $\boldsymbol{x}$. Matrices are in bold capitals, like $\boldsymbol{Z}$ or $\boldsymbol{\Theta}$. Random variables, like $U$, are in non–bold capitals even if they yield vectors. Graph–related objects appear in sans serif, as in D or j. Non–standard sets are in calligraphic capitals, such as $\mathcal{R}$. Components of vectors and elements of matrices retain their bold typeface and their case, for example, $\boldsymbol{z}_i$ or $\boldsymbol{\Theta}_{ij}$.

**Graph terminology.**  To fix terminology, a directed graph, or digraph, $\mathsf{G} = (\mathsf{V}, \mathsf{E})$ comprises a set of nodes $\mathsf{V}$ and a set of edges $\mathsf{E} \subset \mathsf{V} \times \mathsf{V}$. An edge $\mathsf{e} = (\mathsf{i}, \mathsf{j}) \in \mathsf{E}$ may be represented as either $\mathsf{i} \longrightarrow \mathsf{j}$ or $\mathsf{j} \longleftarrow \mathsf{i}$. We do not allow self–loops $\mathsf{i} \longrightarrow \mathsf{i}$. A directed cycle is an aligned sequence of edges $\mathsf{C} = \mathsf{i}_0 \longrightarrow \mathsf{i}_1 \longrightarrow \mathsf{i}_2 \longrightarrow \cdots \longrightarrow \mathsf{i}_{c-1} \longrightarrow \mathsf{i}_0$, for some positive integer, $c$.

A directed acyclic graph (DAG) is a digraph with no directed cycles. A partially–directed graph extends the concept of a digraph to allow undirected edges $\mathsf{i} \longrightarrow \mathsf{j}$. A partially–directed acyclic graph (PDAG) is a partially–directed graph with no *directed* cycles: cycles including undirected edges are allowed [see 16, p. 82].

A Bayesian network, $M = (\mathsf{D}, U)$ consists of a DAG $\mathsf{D} = (\mathsf{V}, \mathsf{E})$ and a vector of random variables indexed by the nodes $U = (U_\mathsf{j})_{\mathsf{j} \in \mathsf{V}}$, such that,

$$p(U) = \prod_{\mathsf{j} \in \mathsf{V}} p(U_\mathsf{j} \,|\, \{U_\mathsf{i} : \mathsf{i} \longrightarrow \mathsf{j} \in \mathsf{E}\}), \tag{6}$$

so the probability for $U_\mathsf{j}$ at a node $\mathsf{j}$ only directly depends on the values of $U$ at the $\mathsf{j}$'s 'parent' nodes.

# B  Discrete backpropagation

**Sampling.**  In the forward pass, via eq. (1), DAG-DB generates i.i.d. P&M samples $\boldsymbol{Z}^{(s)} = \mathrm{MAP}(\boldsymbol{\Theta} + \tau \boldsymbol{\Psi}^{(s)})$, $s = 1, ..., S$, of the binary adjacency matrix, where $\tau$ is the exponential family distribution temperature from eq. (2). For each matrix element ij, the noise samples $\boldsymbol{\Psi}_{ij}^{(s)}$ are generated by the standard logistic distribution. DAG-DB then uses a method such as STE or I-MLE to backpropagate the gradient of resulting empirical loss function, $L = S^{-1} \sum_{s=1}^{S} \ell_{\boldsymbol{Z}^{(s)}}$, from the samples $\boldsymbol{Z}^{(s)}$ to $\boldsymbol{\Theta}$.

Blackbox estimation (BB) [31] can be considered as a special case of I-MLE: with no P&M sampling, it considers a purely deterministic function from $\boldsymbol{\Theta}$ to $\boldsymbol{Z} = \mathrm{MAP}(\boldsymbol{\Theta})$ [1]. BB's inability to deliver good results suggests that the sampling element of DAG-DB plays a useful role. Tracking $\boldsymbol{\Theta}$ components shows that $p(\boldsymbol{Z}; \boldsymbol{\Theta})$ becomes more concentrated around its MAP value as training develops: an initial, broad distribution helps exploration, and, as training progresses, the distribution concentrates around the predicted digraph $\boldsymbol{Z}$.

The maximum size constraint shows the versatility of P&M. With no maximum size constraint, P&M logistic noise sampling is the same as independent Bernoulli sampling for each (non–diagonal) component $\boldsymbol{Z}_{ij}$ [see, for example, 34]. With such a constraint, Bernoulli sampling is not possible because the $\boldsymbol{Z}_{ij}$ components are no longer independent.

**Straight–through estimation (STE).**  Our STE approximation for backpropagation is given by

$$\nabla_{\boldsymbol{\Theta}} L \approx \frac{1}{\tau S} \sum_{s=1}^{S} \nabla_{\boldsymbol{Z}^{(s)}} L. \tag{7}$$

This is as in the original treatments of STE [28, 29], summed over the samples and with an arbitrary choice of proportionality constants.

**Implicit maximum likelihood estimation (I-MLE).**  I-MLE [1] uses a procedure due to Domke [32] to set target distribution parameters, which, for DAG-DB, are, for $s = 1, .., S$,

$$\boldsymbol{\Theta}^{(s)} = \boldsymbol{\Theta} - \lambda \nabla_{\boldsymbol{Z}^{(s)}} L, \tag{8}$$

where $\lambda$ arises from the Domke procedure. In I-MLE, $\lambda$ is treated a hyperparameter, with the approximation of the loss gradient with respect to $\boldsymbol{\Theta}$ then being

$$\nabla_{\boldsymbol{\Theta}} L \approx \frac{1}{\lambda \tau S} \sum_{s=1}^{S} \left[ \mathrm{MAP}(\boldsymbol{\Theta} + \tau \boldsymbol{\Psi}^{(s)}) - \mathrm{MAP}(\boldsymbol{\Theta}^{(s)} + \tau \boldsymbol{\Psi}^{(s)}) \right]. \qquad (9)$$

## C  Details of graphification

Each data point is combined with every sample $\boldsymbol{Z} = \boldsymbol{Z}^{(s)}$ by the 'graphification'

$$(\boldsymbol{x} \diamond \boldsymbol{Z})_{\mathrm{ij}} := x_{\mathrm{i}} \boldsymbol{Z}_{\mathrm{ij}}. \qquad (10)$$

The resulting matrix is then fed into a function $f_{\boldsymbol{\Phi}}$, where the $\boldsymbol{\Phi}$ are learnable parameters. In principle $f_{\boldsymbol{\Phi}}$ could be a multi–layer neural network, for predictions on non–linear Bayesian networks, but, for simplicity, we confine attention to a linear function with no bias, having $\boldsymbol{\Phi} \in \mathcal{R}$ as real–matrix parameters, and

$$[f_{\boldsymbol{\Phi}}(\boldsymbol{M})]_{\mathrm{j}} := \sum_{\mathrm{i}} \boldsymbol{\Phi}_{\mathrm{ij}} \, \boldsymbol{M}_{\mathrm{ij}}, \quad \text{(with no sum over j),} \qquad (11)$$

for a $d \times d$ matrix $\boldsymbol{M}$. Given the digraph $\boldsymbol{Z}$, define the parents of node j as $\mathrm{pa}_{\boldsymbol{Z}}(\mathrm{j}) := \{\mathrm{i} : \boldsymbol{Z}_{\mathrm{ij}} = 1\}$. Equations (10) and (11) then together imply a prediction for the value of $\boldsymbol{x}$ at node j from the values of $\boldsymbol{x}$ its parent nodes,

$$\tilde{x}_{\mathrm{j}} := [f_{\boldsymbol{\Phi}}(\boldsymbol{x} \diamond \boldsymbol{Z})]_{\mathrm{j}} = \sum_{\mathrm{i} \in \mathrm{pa}_{\boldsymbol{Z}}(\mathrm{j})} x_{\mathrm{i}} \, \boldsymbol{\Phi}_{\mathrm{ij}}, \quad \text{(with no sum over j).} \qquad (12)$$

This ensures that, in prediction, a node j's value will only be a function of values associated with its parent nodes.

## D  Details of experiments

### D.1  Synthetic DAGs and data

**Summary.**  Synthetic data is often used for learning hyperparameters and testing as it can be easily and cheaply generated. We follow the approach in the NOTEARS paper [2], also adopted by GOLEM [8] and many other papers. The approach is to generate a random DAG, based on an underlying distribution for randomly–generating graphs or digraphs, and then to randomly–generate an appropriate linear additive noise model (LANM) on that DAG. The LANM is then used to create the synthetic dataset, the $n \times d$ matrix $\boldsymbol{X}$. To generate synthetic DAGs and data, we used open–source code from the GOLEM paper [41], itself derived from NOTEARS [42].

**Generating DAGs.**  The NOTEARS and GOLEM papers consider two types of randomly–generated DAGs: Erdös-Rényi ER$k$ DAGs, where the number of edges is binomially distributed, and in expectation, there are $dk$ edges; and Barabási-Albert SF$k$ DAGs, where the number of edges is fixed, at $k[d - (k+1)/2]$, with some nodes being preferred in terms of having more edges. We use both these types in our experiments.

**Generating data.**  We use a standard Gaussian equal–variance linear additive noise model. Such a model, with variance $\sigma^2 \in \mathbb{R}^+$, is a Bayesian network $M = (\mathrm{D}, U)$ with values $\boldsymbol{u} \sim U$ having each node–component $\boldsymbol{u}_{\mathrm{j}} \in \mathbb{R}$, and satisfying

$$\boldsymbol{u}_{\mathrm{j}} = \left( \sum_{\mathrm{i} \in \mathrm{pa}(\mathrm{j})} \boldsymbol{\Phi}_{\mathrm{ij}} \, \boldsymbol{u}_{\mathrm{i}} \right) + \boldsymbol{\nu}_{\mathrm{j}}, \qquad \boldsymbol{\nu}_{\mathrm{j}} \sim \mathcal{N}(0, \sigma^2) \text{ i.i.d.,} \qquad (13)$$

for some weights matrix $\boldsymbol{\Phi}$. The NOTEARS/GOLEM implementation, which we use, creates such a model by sampling $\boldsymbol{\Phi}_{\mathrm{ij}}$ i.i.d. uniformly randomly from $[-2, -0.5] \cup [0.5, 2]$.

**Potential limitations.** There have been criticisms of this approach. In 2008, it was noted that a similar approach to DAG generation does not generate DAGs uniformly [43, s. 3.3], and this issue also affects the current DAG–generation method. It has also, more recently, been pointed out that linear additive noise models offer clues to prediction methods which may be undesirable. These clues are tendencies for variances and co–variances between nodes to increase as we follow the DAG's edges [44]. Nonetheless these approaches remain the standard synthetic data benchmark and we retain their use, supplemented by our real data experiment, which may mitigate these concerns.

## D.2 Metrics

**CPDAGs.** Different DAGs may represent the same Bayesian network, being, in general, impossible to distinguish simply by observation — that is, by i.i.d. sampling of the Bayesian network's random variable. The Markov Equivalence Class (MEC) of a DAG $D$ is formed by the DAGs $D'$ for which there exists a Bayesian network which may be represented by both $D$ and $D'$. A MEC may be represented by the DAG's class partially–directed acyclic graph (CPDAG) in which some of the DAG's edges become undirected. All DAGs in a MEC have the same skeleton: that is, the same edges, ignoring direction. Any MEC can be represented by a CPDAG: undirected edges indicate that the direction may differ between DAGs within the MEC.

**Identifiability** For some particular Bayesian networks, however, the exact DAG can be identified. Gaussian equal–variance linear noise models have this property. For real data, such as the Sachs data we experiment on, there is no such guarantee. For consistency, we will therefore use metrics based on CPDAGs. More discussion of identifiability can be found in Peters, Janzing & Schölkopf's textbook [16, sec. 7.1.2–4].

**Structural Hamming distance (SHD).** Suppose now that we have a true DAG $D_{\text{true}} = (V, E_{\text{true}})$ and a predicted DAG $D_{\text{pred}} = (V, E_{\text{pred}})$, on the same set of nodes $V$, but with potentially differing edge sets $E_{\text{true}}$ and $E_{\text{pred}}$. A given unordered pair of nodes $\{i, j\}$'s join status will be one of the following: unjoined, joined $i \longrightarrow j$, or joined $i \longleftarrow j$. If we also allow undirected edges, as in CPDAGs, then this adds a fourth possibility $i \longrightarrow j$. SHD is the count of the ordered pairs $i, j$ which differ in terms of join status between $D_{\text{true}}$ and $D_{\text{pred}}$. The class structural Hamming distance $\text{SHD}_c$ is the corresponding metric for the CPDAGs associated with $D_{\text{true}}$ and $D_{\text{pred}}$. Dividing by the number of nodes gives the normalized class structural Hamming distance $\text{nSHD}_c$. (n)SHD is found in, for example, the NOTEARS [2] and GOLEM [8] papers, and $(\text{n})\text{SHD}_c$ in the GOLEM paper. Note that if we predicted an empty DAG $D_{\text{pred}}$, with no edges, we would have $\text{nSHD}_c = |E_{\text{true}}|/|V|$, because a CPDAG has the same skeleton, and so the same number of edges, as any DAG it represents.

**Precision and recall.** In the current context, precision is defined as

$$\text{prec}(D_{\text{true}}, D_{\text{pred}}) = \frac{|E_{\text{pred}} \cap E_{\text{true}}|}{\max(1, |E_{\text{pred}}|)}, \tag{14}$$

treating the edges $i \longrightarrow j$ and $i \longleftarrow j$ as distinct. Similarly, recall is defined as

$$\text{rec}(D_{\text{true}}, D_{\text{pred}}) = \frac{|E_{\text{pred}} \cap E_{\text{true}}|}{\max(1, |E_{\text{true}}|)}. \tag{15}$$

In graph literature [e.g. 2, 8], precision may be replaced by the false discovery rate, which is one minus the precision, and recall may be termed the true positive rate. As for SHD, we may also use the CPDAGs associated with $D_{\text{true}}$ and $D_{\text{pred}}$ to define the class precision $\text{prec}_c$ and class recall $\text{rec}_c$. In this extension to CPDAGs, we regard $i \longrightarrow j$ as a further distinct type of edge between $i$ and $j$.

## D.3 Hyperparameters

**Hyperparameter settings.** To choose hyperparameters, we sought to optimise on a randomly–generated set of six synthetic Erdös-Rényi 'ER2' DAGs with $d = 30$ nodes and a mean of 60 edges. On each of these graphs, a synthetic dataset was created using a Gaussian equal–variance linear additive noise model. We assessed the viability of I-MLE, STE, SFE and BB as the discrete backpropagation method, finding I-MLE and STE to be the most promising. In exploring hyperparameter settings, we used a combination of Optuna [45, 46], a well–known package for Bayesian

Table 2: Hyperparameters for DAG-DB.

| Symbol | Description | STE_84 | IMLE_None |
|---|---|---|---|
| **Basic set–up** | | | |
| $n$ | Total number of data points in data matrix $\mathbf{X}$ | 1000 | |
| | Number of epochs training | 1000 | |
| | Whether to shuffle batches for each new epoch | True | |
| | Batch size | 16 | 8 |
| **Discrete backprop** | | | |
| | Method for backpropagation from discrete $\mathbf{Z}$ to continuous $\mathbf{\Theta}$ | STE | I-MLE |
| $S$ | Number of P&M samples | 10 | 47 |
| $\tau$ | Temperature in the exponential family distribution of eq. (2) | $1.771 \times 10^{-1}$ | $8.786 \times 10^{-1}$ |
| $\lambda$ | I-MLE Domke hyperparameter | n/a | $2.714 \times 10^{1}$ |
| | Width of initial uniform distribution for each $\mathbf{\Theta}$ component. (Note that $\mathbf{\Phi}$ is initialised as for `torch.nn.Linear` layers) | $2.169 \times 10^{-1}$ | $1.137 \times 10^{-4}$ |
| **Optimization** | | | |
| | Optimizer | `torch.optim.Adam`, as in I-MLE [1] | |
| | Learning rate for $\mathbf{\Theta}$ | $1.134 \times 10^{-4}$ | $1.616 \times 10^{-3}$ |
| | Learning rate for $\mathbf{\Phi}$ in $f_{\mathbf{\Phi}}$ | $1.232 \times 10^{-2}$ | $3.720 \times 10^{-1}$ |
| **Regularization** | | | |
| $\rho_{\mathrm{DAG}}$ | Coefficient for the $r_{\mathrm{DAG}}$ regularizer | $4.101 \times 10^{-1}$ | $1.575 \times 10^{-1}$ |
| $\rho_{\mathrm{sp}}$ | Coefficient for the $r_{\mathrm{sp}}$ regularizer | $1.023 \times 10^{-2}$ | $1.208 \times 10^{-3}$ |
| $M$ | Maximum size of digraph $\mathbf{Z}$ | 84 | n/a |

optimisation of hyperparameters, and manual adjustment, for example taking an Optuna–generated set of hyperparameters and varying the standard probability distribution used as noise for P&M. We also performed a grid search over hyperparameters, which was less successful in identifying optimal settings. We identified two settings as particularly promising, STE_84, using STE with a maximum size constraint of 84 edges, and IMLE_None, using I-MLE with no maximum size constraint.

We chose synthetic ER2 DAGs with 30 nodes, as lying roughly in the middle of the range of types and node–numbers considered here, and in the NOTEARS and GOLEM papers. Note that, following appendix D.2, a DAG with no edges would score an expected $\mathrm{nSHD_c} = 2$ when compared with a 'true' ER2 DAG, suggesting a cut–off maximum $\mathrm{nSHD_c}$. With BB, we did not find any settings which delivered $\mathrm{nSHD_c} < 2$; whilst for SFE, we only managed $\mathrm{nSHD_c}$ slightly less than 2. Table 2 shows the best hyperparameters for DAG-DB we found for STE_84 and IMLE_None.

### D.4 Ablation of regularizers

We can see from tables 3 and 4 that, for both STE_84 and IMLE_None settings, all ablations worsen the metrics, although, for IMLE_None , the effect of ablating $r_{\mathrm{sp}}$ is very minor, as might be expected from table 2 which shows that its coefficient $\rho_{\mathrm{sp}}$ is rather small.

Table 3: Effects of ablation for `STE_84`, with crosses indicating ablation of the relevant regularizer.

| $\texttt{MAX\_SIZE} = 84$ | ✗ | ✗ | ✗ | ✗ | ✓ | ✓ | ✓ | ✓ |
|---|---|---|---|---|---|---|---|---|
| $r_{\mathrm{DAG}}$ | ✗ | ✗ | ✓ | ✓ | ✗ | ✗ | ✓ | ✓ |
| $r_{\mathrm{sp}}$ | ✗ | ✓ | ✗ | ✓ | ✗ | ✓ | ✗ | ✓ |
| $\mathrm{nSHD_c}$ | 3.901 | 1.022 | 2.068 | 1.708 | 1.406 | 1.286 | 0.919 | 0.732 |
| $\mathrm{prec_c}$ | 0.282 | 0.724 | 0.481 | 0.552 | 0.565 | 0.643 | 0.719 | 0.814 |

Table 4: Effects of ablation for `IMLE_None`. Crosses are as in table 3, but note that, in this table, $\texttt{MAX\_SIZE} = \texttt{None}$ is marked by ✓, while 'ablation' to $\texttt{MAX\_SIZE} = 66$, which was found to be a useful maximum size for I-MLE, is marked by ✗.

| $\texttt{MAX\_SIZE} = \texttt{None}$ | ✗ | ✗ | ✗ | ✗ | ✓ | ✓ | ✓ | ✓ |
|---|---|---|---|---|---|---|---|---|
| $r_{\mathrm{DAG}}$ | ✗ | ✗ | ✓ | ✓ | ✗ | ✗ | ✓ | ✓ |
| $r_{\mathrm{sp}}$ | ✗ | ✓ | ✗ | ✓ | ✗ | ✓ | ✗ | ✓ |
| $\mathrm{nSHD_c}$ | 1.425 | 1.411 | 1.268 | 1.293 | 1.401 | 1.311 | 1.064 | 1.058 |
| $\mathrm{prec_c}$ | 0.584 | 0.589 | 0.637 | 0.631 | 0.585 | 0.619 | 0.733 | 0.736 |

## D.5 Variation by DAG

The standard box–and–whisker plots of fig. 2 indicate variation over 24 test DAGs. Each box covers from the first to the third quartile, with an interior line indicating the median. A white dot indicates the mean. The whiskers each extend to cover all points within a further 1.5 times the inter–quartile range, whilst any points outside that extended range are marked individually.

We also considered whether 24 DAGs was enough to give a reasonable sample, experimenting via a test set of 240 ER2 DAGs with 30 nodes. We found that taking a $100\,000$ random 24 DAG sub–samples of the 240 gave standard errors for $\mathrm{nSHD_c}$ of 0.1 for `STE_84` and 0.15 for `IMLE_None`, and of 0.03 for $\mathrm{prec_c}$ with either method. We then examined how much results could vary in repeated prediction on the same 24 DAGs. We made predictions ten times for a constant set of 24 DAGs, finding variation from minimum to maximum for $\mathrm{nSHD_c}$ of 0.03 for `STE_84` and 0.05 for `IMLE_None`, and of 0.01 for $\mathrm{prec_c}$ with either method. These results suggest that 24 DAGs may be a sufficient sample size for testing our methods.

## D.6 Methods used for comparison

We follow the methods used for comparison in the GOLEM paper [8], with one exception. The combinatoric methods we use are FGES [21] (formerly known as FGS [47]) and PC [19]. The GOLEM paper used the Conservative PC algorithm [48] as its version of PC, however, this produces results in a form more general than a CPDAG, complicating comparisons. We therefore use the PC-Stable [20] variant, which, like the original PC algorithm, returns a CPDAG. The continuous methods we compare with are NOTEARS [2] and GOLEM [8]. For NOTEARS, as was done in the GOLEM paper, we employ the NOTEARS-L1 variant with the settings given as defaults for linear SEMs in the NOTEARS code repository. GOLEM provides two variants: GOLEM-EV, suitable for data generated with equal variance, and GOLEM-NV, suitable for non–equal variance data. Accordingly, we use GOLEM-EV for our synthetic experiments and GOLEM-NV for our real data experiment.

To run experiments for these methods, we used `py-causal` [49] for FGES and PC, and the NOTEARS [42] and GOLEM [41] repos. Licences for these packages, and for the code we adapted for GFAS, are shown in table 5.

## D.7 Supplementary results on Sachs experiments

See table 6 for results from I-MLE, with training using digraphs, `IMLE_None`, and with training using DAGs, `IMLE_None_Tr`. Table 7 shows standard deviations, over 24 runs, for these methods. The

Table 5: Licences for comparison methods and GFAS.

| Code | Ref. | Link to licence description |
|------|------|-----------------------------|
| `py-causal` | [49] | GNU Lesser General Public License v2.1+ |
| NOTEARS | [42] | Apache License 2.0 |
| GOLEM | [41] | Apache License 2.0 |
| GFAS | [39] | BSD License |

Table 6: As for table 1, but comparing variants with and without DAGs in training for a range of values of $\lambda$. For each $\lambda$, the first row is the mean and the second row is the median. $\lambda = 27.14$ is the default setting for `IMLE_None`.

| | IMLE_None | | | | IMLE_None_Tr | | | |
|---|---|---|---|---|---|---|---|---|
| $\lambda$ | $SHD_c$ | $prec_c$ | $rec_c$ | pred. size | $SHD_c$ | $prec_c$ | $rec_c$ | pred. size |
| 0.01 | 12.0 | 1.000 | 0.294 | 5.0 | 13.3 | 0.778 | 0.314 | 7.0 |
|      | 12   | 1.000 | 0.294 | 5   | 13.0 | 0.750 | 0.294 | 7   |
| 0.1  | 12.5 | 0.950 | 0.267 | 4.8 | 12.3 | 0.950 | 0.275 | 4.9 |
|      | 12   | 1.000 | 0.294 | 5   | 12   | 1.000 | 0.294 | 5   |
| 1.0  | 12.3 | 0.823 | 0.275 | 5.7 | 13.6 | 0.584 | 0.208 | 6.1 |
|      | 12.5 | 1.000 | 0.265 | 6   | 14   | 0.536 | 0.206 | 6   |
| 10   | 12.1 | 0.894 | 0.287 | 5.5 | 13.2 | 0.520 | 0.292 | 9.7 |
|      | 12   | 1.000 | 0.294 | 6   | 13   | 0.500 | 0.294 | 9.5 |
| 27.14 | 12.7 | 0.869 | 0.255 | 5.0 | 13.8 | 0.475 | 0.287 | 10.5 |
|       | 13   | 1.000 | 0.235 | 5   | 14   | 0.455 | 0.294 | 11   |
| 100  | 12.1 | 0.894 | 0.287 | 5.5 | 14.0 | 0.451 | 0.279 | 10.8 |
|      | 12   | 1.000 | 0.294 | 5   | 14   | 0.455 | 0.294 | 10.5 |

Table 7: As for table 6, but showing standard deviations over the 24 runs.

| | IMLE_None | | | | IMLE_None_Tr | | | |
|---|---|---|---|---|---|---|---|---|
| $\lambda$ | $SHD_c$ | $prec_c$ | $rec_c$ | pred. size | $SHD_c$ | $prec_c$ | $rec_c$ | pred. size |
| 0.01[a] | 0.0 | 0.000 | 0.000 | 0.0 | 1.0 | 0.113 | 0.028 | 1.2 |
| 0.1     | 0.8 | 0.135 | 0.046 | 0.5 | 1.0 | 0.135 | 0.059 | 0.8 |
| 1.0     | 1.3 | 0.198 | 0.077 | 0.8 | 1.1 | 0.175 | 0.057 | 0.9 |
| 10.0    | 1.3 | 0.169 | 0.076 | 1.0 | 0.6 | 0.061 | 0.012 | 1.2 |
| 27.14   | 1.2 | 0.189 | 0.073 | 0.9 | 0.8 | 0.074 | 0.020 | 1.5 |
| 100.0   | 1.2 | 0.186 | 0.073 | 0.6 | 0.9 | 0.076 | 0.031 | 1.8 |

[a] Our data reports confirm that the zero standard deviation results for `IMLE_None` are indeed distinct runs.

varied hyperparameter, $\lambda$, is the Domke hyperparameter discussed in appendix B. We varied this as, in I-MLE, results can be particularly sensitive to $\lambda$ [1]. Table 7's $\lambda = 27.14$ rows for `IMLE_None` are the standard deviations corresponding to table 1. GOLEM and NOTEARS results have very small standard deviations, whilst FGES and PC are deterministic.

As noted, DAG training is time consuming. Running time for each single prediction increases from about 0h15m with `IMLE_None` to around 1h15m with `IMLE_None_Tr`.

### D.8 Computational resources

Computations were done on a combination of a laptop with a GeForce RTX 3080 Mobile 16GB GPU and a computer cluster. Prediction for a single ER2 graph on 30 nodes (roughly the mid–range of the graphs shown in fig. 2) took around 0h12 on the laptop, for the longest–running of all the methods in fig. 2, `IMLE_None`, giving around 72 hours to compile that element of the figure. The corresponding `STE_84` computation took around a quarter of that time, and the other methods we compared with were quicker, giving a total time to compute fig. 2 of the order of 100 hours. Hyperparameter exploration, ablation experiments and Sachs experiments are estimated to add another 200 hours compute, while the experiments on variation described in appendix D.5 took around 120 hours.

