# OpenReview forum: "Learning Discrete Directed Acyclic Graphs Via Backpropagation"
_NeurIPS.cc/2022/Workshop/nCSI — nCSI WS @ NeurIPS 2022 Poster_

### Official Review · Reviewer_AhCw · 2022-10-05
**Marginal contribution, performance unimpressive**

**Rating:** 2
**Confidence:** 2

**Review:**

Summary:
The authors attempt to learn sparse DAGs that explain observational data. In contrast with prior work that considers relaxing the set of graphs to a real space, the authors propose to learn a distribution over graphs via backprop methods on discrete sets. This distribution is learned jointly with a model (here considered linear) that generates the data according to the graph.
The loss objective is reconstruction and L1 and NOTEARS regulizers for sparsity and acyclicity.
For optimizing the distribution over graphs, the authors consider the straight through estimator and the Implicit-MLE method.

Strength:
- Gives some insight into which methods do and do not work for this problem.

Weaknesses:
- The contribution is marginal. The authors experimented with some pre-existing discrete backprop methods on a problem that has previously been addressed by relaxation methods.
- The proposed method performs worse than the relaxation methods.
- The authors don't really motivate why relaxation methods are an insufficient/undesirable solution to the problem.
- Many important details relegated to the appendix, making the paper hard to read.

Conclusion:
I recommend a borderline accept for this paper. Even though it has some flaws, it may still be interesting for the community to learn which approaches do and do not work on this problem.

---

### Official Review · Reviewer_Hgbd · 2022-10-10
**The proposed method of learning DAGs using discrete backpropogation is novel and innovative.**

**Rating:** 3
**Confidence:** 2

**Review:**

The proposed method of learning DAGs using discrete backpropogation is novel and innovative. This paper is well-thought-out in terms of both conceptual and implementation.
The discussed experiments appear to be adequate to support the research work. Figures, results, and tables provide a solid context. The related work is properly referenced and contextualized.

Suggestions: \
The method section is well-explained, but the authors could simplify it if they so desired. To better connect with the main section, the conclusion should be expanded and improved.
Rewriting the experiments section can help to justify the strong idea and referenced results
The paper flow is excellent. However, the grammar and acronyms could be better. Some reference pointers are missing.

---

### Meta-Review · Area_Chair_ZGrZ · 2022-10-19

**Recommendation:** 2
**Confidence:** 1

**Metareview:**

This paper presents a backpropagation-based approach for learning DAGs, and follow up with some comprehensive results. I’m not an expert in the area, but R2 seems to suggest that the contribution is marginal, and there is insufficient explanation for when other methods perform poorly. I’d recommend getting another review if possible, by someone who knows the area. If technically sound, perhaps this is a weak accept.

---

### Decision · Program_Chairs · 2022-10-20

Accept (Poster)